# Use of Antibiotics following Snakebite in the Era of Antimicrobial Stewardship

**DOI:** 10.3390/toxins16010037

**Published:** 2024-01-11

**Authors:** Helena Brenes-Chacon, José María Gutiérrez, María L. Avila-Aguero

**Affiliations:** 1Pediatric Infectious Diseases Division, Hospital Nacional de Niños “Dr. Carlos Sáenz Herrera”, Centro de Ciencias Médicas, Caja Costarricense de Seguro Social (CCSS), San José 10103, Costa Rica; helena.brenes@ucr.ac.cr; 2Instituto Clodomiro Picado, Facultad de Microbiología, Universidad de Costa Rica, San José 11501, Costa Rica; jose.gutierrez@ucr.ac.cr; 3Escuela de Medicina, Universidad de Ciencias Médicas (UCIMED), San José 10108, Costa Rica; 4Affiliated Researcher, Center for Infectious Disease Modeling and Analysis (CIDMA), Yale University New Haven, New Haven, CT 06520, USA

**Keywords:** snakebite envenoming, severe envenoming, wound infection, antibiotic prophylaxis

## Abstract

Even though there are guidelines for the management of snakebite envenoming (SBE), the use of antibiotics in this pathology remains controversial. The aim of this study is to provide a narrative review of the literature and recommendations based on the best available evidence regarding antibiotic use in SBE. We performed a narrative review of relevant literature regarding SBE and antibiotic use as prophylaxis or treatment. A total of 26 articles were included. There is wide use of antibiotics in SBE; nevertheless, infection was not necessarily documented. The antibiotics used varied according to the study, from beta lactams to lincosamide and nitroimidazoles, and from monotherapy to combined antimicrobials. The most common recommendations were to manage skin and soft tissue infections and avoid infectious complications, but these suggestions are not necessarily based on bacteriological findings. Prophylactic use of antibiotics in SBE is discouraged in most studies. Antibiotic prescription in SBE should be based on the susceptibility of microorganisms isolated from the affected tissue or identified in snakes’ oral cavities. Antibiotics should be reserved only for patients with a demonstrated infection, or those at a high risk of developing an infection, i.e., presenting severe local envenoming, local signs of infection, or those with incorrect manipulation of wounds. Prospective studies are needed to correlate microbiological findings at the wound site and the response to antibiotic use.

## 1. Introduction

Snakebite envenoming (SBE) is a neglected tropical disease (NTD) responsible for high morbidity and mortality. There are more than 250 species of venomous snakes worldwide that are considered medically important by the World Health Organization (WHO) [1]. More than 5.8 billion people are at risk of encountering a venomous snake, and each year about 2.7 million cases are reported, resulting in 81,000–138,000 deaths [1].

SBE disproportionately affects children in low-income settings [2,3], often leading to permanent physical and psychological sequelae [4]. Due to their smaller size and lower volumes of distribution related to the injected venom, children often present with more severe envenoming, associated with more rapid development of neurotoxicity, coagulopathy, and severe local tissue damage [2,5].

Bacterial infections are a secondary complication of wounds caused by animal bites, including those inflicted by snakes [6,7]. The pathogenic microorganisms causing an infection are not only the ones from the patient’s skin flora but also those present in the snake’s oral cavity. Several studies have isolated bacteria from the oral cavity and venom of several species of snakes, which are likely to be involved in infections in cases of SBE [8,9]. Several studies have shown that bacterial infections are commonly observed in SBE inflicted by a variety of viperid and elapid species in different geographical settings [6,9,10].

Despite the relevance of infectious complications, the burden of infection in snakebites remains largely unknown, and reports tend to show variable findings. Infection rates range from 9 to 77% of patients [11,12], with data in children often being limited and extrapolated from adults. Inappropriate first-aid interventions, such as the use of tourniquets, local application of chemicals or natural products, electric shocks, and incisions at the bite site, among others, are likely to increase the risk of infection [13,14,15].

However, even though snakebites have been shown to have the potential to cause primary infections via the inoculation of infectious agents present in the venom and oral cavity of snakes, and secondary infections as a result of extensive tissue damage and bacterial superinfection, there is no consensus or specific guidelines regarding the use of antibiotics to treat these infections. In many instances, they are used prophylactically or without documenting the occurrence of infection. These antibiotics are often used as initial empirical therapy for many infectious diseases, so their use must be carefully considered.

The objective of this study is to carry out a narrative review of the literature on this topic and provide recommendations based on the best available evidence, which can be applied in centers that manage patients suffering with SBE.

## 2. Results

From 1980 to 2023, we identified twenty-six publications focused on snakebite and antibiotic use from a range of countries with high incidence of SBE (Figure 1 and Table 1). The description of wound infection associated with SBE is mentioned in many articles, and a considerable percentage of patients developed this complication. Soft tissue infection such as cellulitis or abscess formation was described in 10% to 25% of patients in most studies [6,9,16,17,18,19,20,21,22], but its occurrence could be higher depending on the severity of envenoming [23,24]. In contrast, in some settings the prevalence of infections in SBE patients is lower [25,26,27], probably related to the type of envenoming and management received by patients in these settings.

Risk factors for wound or soft tissue infection were studied in envenomings caused by diverse species of the families Viperidae and Elapidae [7,17]. Although many factors are involved in the development of complications secondary to SBE, a consistently higher incidence of infection was described in patients with clinically moderate to severe envenoming [22], including cases with necrosis [38]. Necrosis, which is associated with tissue damage in envenomings by species of the family Viperidae and some species of the family Elapidae, favors the presence of bacterial infection. Houcke et. al., in their studies in French Guiana, where *Bothrops atrox* is responsible for most bites, identified necrosis as an independent factor associated with infection in these envenomings (OR 13.15, CI: 4.04–42.84, *p* < 0.001), along with thrombocytopenia, and rhabdomyolysis [38]. There are other described risk factors for infection, such as self-manipulation of the wound prior to receiving medical attention [34,41,42], envenoming caused by species of *Bothrops* sp. and by elapid species of the genus *Naja* that can induce significant tissue damage [7,19,38], or a delay in medical care after the bite [6]. A study in Brazil identified several laboratory parameters that correlated with a higher risk of infection, such as elevated concentration of fibrinogen, alanine aminotransferase (ALT), and C-reactive protein (CRP) [33]. Some of the most common organisms described in the literature as causes of infection in snakebite envenoming are *M. morganii*, *Proteus* sp., *S. aureus*, *Enterococcus* sp., *A. hydrophila*, and *E. coli* (Table 1).

We focused our search on two main aspects: first, the antibiotics that were used to treat SBE cases, and second, the indication of antibiotic use and recommendations. Use of antibiotics in SBE was reported from 12% to as high as 100% of affected patients (Table 1) [36,43,44,45]. They were used either prophylactically or in patients with suspected or confirmed infection. Many antibiotics have been used, including beta-lactams (such as penicillin, amoxicillin–clavulanate, piperacillin–tazobactam, and cephalosporins), aminoglycosides (such as gentamicin and amikacin), nitroimidazoles (metronidazole), lincomycin (clindamycin), or quinolones (Table 1). Older studies included chloramphenicol in some settings [16,21,28,29]. Interestingly, we identified that a sensitivity analysis for identified bacteria was performed in some studies (17 contained reports of antibiotic sensitivity patterns vs. 9 studies that did not report it). Some groups used antibiotics based on the microorganisms most frequently described by other works, or sometimes, they used the reference of sensitivity patterns from other studies.

We also focused our review on the criteria for recommending antibiotic use. Several studies proposed the prophylactic use of antibiotics in SBE owing to the likelihood of bacterial infections in this pathology [16,32,35]. In contrast, most studies emphasize that the prophylactic use of antibiotics should be avoided, and instead suggest that they should be used only when there is evidence of infection in these patients. Three controlled studies in SBE in South America evaluated the use of prophylactic antibiotics. It was found that the incidence of infection was not reduced in patients receiving antibiotics compared to those who did not receive them [21,29,33]. Other studies also argue against the use of prophylactic antibiotics in SBE (Table 1) [23,25,26,37]. Thus, there is a predominant view in the reviewed literature that the prophylactic administration of antibiotics in SBE is not warranted and that they should be used only when there is clinical or bacteriological evidence of infection.

## 3. Discussion

We identified twenty-six articles that fulfilled the search criteria on the topic of antibiotic use in SBE. The incidence of infection in SBE is highly variable and depends on several factors. Infections might result from the inoculation of bacteria present in the oral cavity and the venom of snakes [8,46], as well as from bacterial superinfection secondary to local tissue damage and disruption of the skin integrity. Identification of risk factors of infection is essential to determine the cases in which the rational use of antibiotics is indicated. The reviewed literature mentions that the risk of wound infection is higher in patients with moderate or severe envenoming, self-manipulation of the wound [25,26,38], bites inflicted by species that cause pronounced local tissue necrosis such as those of the genera *Bothrops* in Latin America [10,47] and cytotoxic *Naja* species [19], or a delay in the access of medical care after the bite [6].

The role of venom-induced local tissue damage, i.e., necrosis and ischemia, as a factor that favors infection, has been demonstrated experimentally [48]. The snake species causing envenoming is also important to consider when suspecting infection. Different snake species predominate in different regions, and variations in the associated pathologies are likely to play a role in the incidence of infections [34,49,50].

In the studies reviewed here, broad-spectrum antibiotics were generally used, either alone or, more often, as combinations, with a predominance of third-generation cephalosporins, ampicillin, metronidazole, clindamycin, and occasionally oral ciprofloxacin (Table 1). There is no consensus regarding which antibiotics to use in SBE, and several studies recommend selecting the antibiotics based on the predominant bacteria of the mouth of snakes [16,30,40,51]. In addition, care should be taken to consider the possible adverse effects of some antibiotics in the context of the pathophysiology of envenoming. For example, in the case of aminoglycosides, their use might be detrimental in the case of neurotoxic envenoming owing to the possible exacerbation of clinical symptoms secondary to the blocking effect at the neuromuscular junctions, and its nephrotoxic side effect. On the other hand, although amoxicillin–clavulanate is recommended for the treatment of soft tissue infections for other animal bites, its use in SBE is controversial, and several studies do not support its use [11,22,33]. Also, the effect of antibiotics in other organs, such as the impact some of them have on renal function, may also be detrimental in a disease in which renal compromise is part of the findings in severe envenoming [52,53].

The routine use of antibiotics as prophylaxis after snakebite has been proposed by some authors and is routinely applied in several hospital settings. However, this practice is controversial, and in most studies analyzed there is a consensus against it, since it is not supported by clinical evidence in controlled trials. Therefore, a rational use of antibiotics is mandatory in every disease associated with infection, given the emergence of multi-resistant bacteria, and SBE is not an exception [54,55]. In ideal conditions, before starting antibiotics, aerobic and anaerobic cultures should be carried out to identify the infecting microorganisms and to select the most effective antibiotics. However, in many rural settings of sub-Saharan Africa, Asia, and Latin America, this may not be possible due to limited resources. Therefore, in many health facilities in regions of high incidence of SBE, the identification of patients that require antibiotic therapy is usually based on clinical evidence of infection, which is often associated with prominent tissue damage as a consequence of envenoming.

Our review has limitations. The literature regarding antibiotics in SBE is heterogeneous, and randomized studies comparing antibiotic use are limited. The use of antibiotics described in the publications was based on standard of care in individual settings, making comparisons difficult. Nevertheless, the description of antibiotics used in different studies show a general picture of the management of infections in snakebite envenoming.

## 4. Conclusions

The use of antibiotics in SBE is a common practice, and in some cases, it is used prophylactically. Although the literature on the subject is heterogenous, there is a growing consensus that antibiotics should not be used in all cases of SBE, and instead they should be reserved only for patients with a demonstrated infection, or those at a high risk of developing an infection, i.e., presenting severe local envenoming, local signs of infection, or those with incorrect manipulation of wounds. Prospective studies need to be conducted to establish the actual incidence of infection in SBE in different settings, to correlate microbiological findings and pathology at the wound site, as well as to select the most effective antibiotic therapy. There is also a need to generate guidelines and conduct prospective studies on this relevant aspect of SBE.

## 5. Materials and Methods

To identify published studies in the field, we reviewed the most relevant literature on this subject. The goal was to gather information regarding SBE and antibiotic use, as well as recommendations for prophylaxis with antibiotics or treatment for established infections. Previous publications on these topics were analyzed in detail, and general trends were identified.

A search for biomedical literature in PubMed, Scopus, Embase, and Cochrane library databases was carried out using the following terms: Snakebite, Snakebite AND antibiotics, Antibiotics in snakebite, Snakebite envenoming AND antibiotics, Snakebite envenomation AND antibiotics. We found a total of 9360 articles, of which 287 specifically discussed antibiotic use in SBE. Abstracts and articles were reviewed by two of the authors (HB-C, MLA-A). We excluded those in languages other than English, Spanish or Portuguese, when no clinical data were included and when only laboratory work was reported. Case reports and small case series were also excluded.

## Figures and Tables

**Figure 1 toxins-16-00037-f001:**
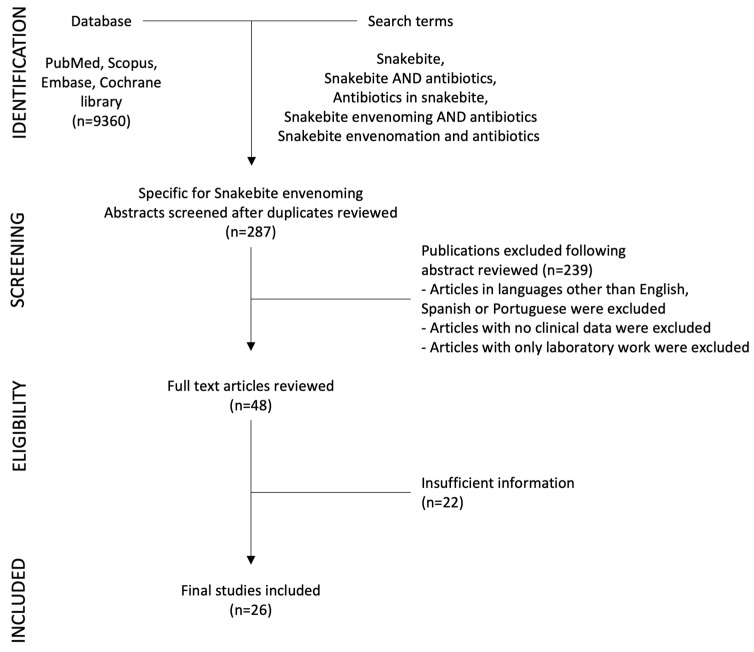
Diagram of selection of studies.

**Table 1 toxins-16-00037-t001:** Included studies with main characteristics, objectives, and principal outcomes and observations.

Title	Author(s)	Year	Objective	Population	Snake Species	Isolated Bacteria	Antibiotic Use Recommendation	Findings and Recommendations
Bacteriology of snakebite abscess	Kerrigan, K [16]	1992	Efficacy of prophylactic antibiotic in reducing incidence of snakebite injuries has never been documented and needs to be studied.	312 patients	Viperidae family	*S. aureus* *Streptococcus* sp.	Gentamicin and chloramphenicol	Abscesses in 9% of patients. Broad spectrum antibiotic prophylaxis should be used based on local flora of snakes in every setting.
The incidence of wound infection following crotalid envenomation	Clark, R et al. [25]	1993	Assess the incidence of wound infection and evaluate the need for prophylactic antibiotics.	54 patients	Crotalid (rattlesnake)	*P. aeruginosa* *Proteus* sp. Coagulase negative *Staphylococcus*, *Clostridium* sp. *B. fragilis*	Amoxicillin/clavulanate, nafcillin Cephalexin, Cefazolin, ceftriaxone	22% used prophylactic antibiotics. 1.8% developed wound infection. Routine use of prophylactic antibiotics may not be warranted.
Microbiological studies of abscesses complicating *Bothrops* snakebite in humans: A prospective Study	Jorge, M et al. [28]	1994	Identify microorganisms responsible for abscess formation at the bite site and antimicrobial sensitivity.	40 patients	*Bothrops* spp.	*M. morganii* *P. rettgeri* *Bacterioides* sp. *Enterobacter* sp. *Streptococcus* sp.	Chloramphenicol Benzyl penicillin + gentamycin	75% presented with abscesses. Chloramphenicol is recommended in cases of infection. No specific recommendation about use of prophylactic antimicrobial treatment is stated.
Antibiotic prophylaxis for pit viper envenomation: Prospective, Controlled Trial	Kerrigan, K [21]	1997	Determine whether prophylactic antibiotics can decrease the incidence of infectious complications at the anatomical site of pit viper envenomation.	114 patients	Viperidae family	*E. coli* *Klebsiella* sp. *Enterobacter* sp. *Proteus* sp. *S. aureus*	Gentamicin and chloramphenicol	7.9% developed abscesses. 67% of patients who developed abscesses received antibiotic prophylaxis. Antibiotics are not indicated as prophylactic therapy for pit viper envenomation. Antibiotics do not prevent infectious complications, are not cost-effective, and may select resistant organisms.
Antibiotics after Rattlesnake envenomation	LoVecchio, F et al. [26]	2002	Describe the incidence of infection following rattlesnake bite.	56 patients	Crotalid (rattlesnake)	No bacteria isolated	Antibiotics used are not specified	5% received antibiotics. No cases of documented infection. Prophylactic antibiotics are not indicated in patients with rattlesnake bites.
Failure of chloramphenicol prophylaxis to reduce the frequency of abscess formation as a complication of envenoming by *Bothrops* snakes in Brazil: A double-blind randomized controlled trial	Jorge, M et al. [29]	2004	Comparison between chloramphenicol and placebo prophylaxis to reduce abscess formation as a complication of *Bothrops* spp. Envenoming.	251 patients	*Bothrops* spp.	*M. morganii* *E. coli*	Chloramphenicol vs. placebo prophylaxis	Abscesses developed in 6 (4.9%) patients with chloramphenicol and 6 (4.7%) in the placebo group. Use of chloramphenicol for snakebite victims with local signs of envenoming is not effective for the prevention of local infection.
Wound infections secondary to snakebite	Garg, A et al. [30]	2009	Evaluate the aerobic bacteria responsible for snakebite-associated wound infection and antibiogram of these isolates.	43 patients	Snake species are not specified	*S. aureus* *E. coli* *Coagulase negative Staphylococcus*	Gentamicin, amikacin, ciprofloxacin, ceftriaxone, meropenem	Antibiotic use should be prescribed according to local susceptibilities.
Bacterial infection in association with snakebite: A 10-year experience in a northern Taiwan medical center	Chen, CM et al. [9]	2011	Survey of patients admitted for snakebites at a medical center in northern Taiwan.	231 patients	*Trimeresurus* spp. *Naja*, *B. multicinctus*	*M. morganii* *Enterococcus* sp. *B. fragilis* *P. aeruginosa*	Amoxicillin–clavulanate, ciprofloxacin, piperacillin–tazobactam	25% developed cellulitis or wound infection. Cobra bite-related injuries were more severe than those inflicted by other species. Prophylactic use of antibiotics is controversial.
Wound infections secondary to snakebite in central Taiwan	Huang, LW et al. [31]	2012	Investigate the treatment of secondary infection following snakebites in Taiwan.	121 patients	*Naja atra*, *T. mucrosquamatus*, *T. stejnegeri*, *B. multicinctus*	*M. morganii* *A. hydrophila* *Enterococcus* sp.	piperacillin-tazobactam, quinolonee, cephalosporins	28% developed wound infection.
Pattern of use of Antibiotics following snakebite in a tertiary care hospital	Palappallil, D et al. [32]	2015	Pattern of antibiotics used following snakebite envenomation in a tertiary care hospital of Kerala.	313 patients	Snake species are not specified	Bacteria identified are not specified	Ampicillin, cloxacillin, metronidazole, cefotaxime, piperacillin–tazobactam, ciprofloxacin	There is a high prescription of antibiotics in snake bitten patients (95%). Clinical outcomes of patients with or without antibiotics were not different.
Bacteriology of *Naja atra* snakebite wound and its implications for antibiotic therapy	Mao, YC et al. [24]	2016	Understand the bacteriology of *N. atra* bite wound.	112 patients	*Naja atra*	Gram-negative rod *M. morganii* *Enterococcus* sp. *Proteus* sp. *A. hydrophila* *Bacterioides* sp.	Ureidopenicillin Aminopenicillin + third-generation cephalosporin or fluoroquinolone	77% developed wound infection, including cellulitis, tissue necrosis, gangrene, and necrotizing fasciitis. Patients may have received antibiotics in the prehospital setting.
Poor efficacy of preemptive amoxicillin clavulanate for preventing secondary infection from *Bothrops* snakebites in the Brazilian Amazon: A randomized controlled clinical trial	Sachett, J et al. [33]	2017	Assess the efficacy of amoxicillin clavulanate for reducing secondary infection. Identify associated factors for secondary infections from snakebites.	186 patients	*Bothrops* spp.	*M. morganii* *S. aureus*	Amoxicillin-clavulanate use is not recommended	Antibiotic schemes suggested for the treatment of secondary infection are not based on good evidence. No evidence that antibiotics decreased risk of associated secondary infection. Higher risk of infection in patients with elevated fibrinogen, ALT, CRP.
Wound infection secondary to snakebite	Wagener, M et al. [17]	2017	Determine the bacterial causation of wound infection secondary to snakebite.	164 patients	Snake species are not specified	*M. morganii* *Proteus* sp. *E. faecalis*	Ceftriaxone, ciprofloxacin, gentamicin, amikacin	26% patients developed infection. Recommendation advises against prophylactic use of antibiotics to treat all snakebites. Use of antibiotics in snakebite is widespread and not directed. Good antibiotic policy is strongly advocated.
Prophylactic antibiotics are not needed following Rattlesnake Bite	August, J et al. [34]	2018	Determine efficacy of prophylactic antibiotics for venomous snakebites in the US.	2748 patients	Crotalid (rattlesnake)	*S. aureus* *E. coli* *Enterococcus* sp. *B. fragilis*	Antibiotics used are not specified	Self-manipulation of wounds is associated with post bite infection. No recommendation in prophylaxis after rattlesnake bites.
Bacterial infections associated with Viperidae snakebites in children: a 14 year experience at the Hospital Nacional de Niños de Costa Rica	Brenes-Chacon, H et al. [6]	2019	Infectious complications associated with Viperidae snakebites in children.	75 patients	Viperidae family	*M. morganii* *A. hydrophila* *P. rettgeri*	Combination of penicillin or clindamycin with an aminoglycoside	19% developed wound infection. Infection complications are more frequent in patients with severe envenomation or patients with delayed medical care. Use of antibiotic prophylaxis is controversial.
Infectious complications following snakebite by *Bothrops Ianceolatus* in Martinique: a case series	Resiere, D et al. [22]	2020	Investigate the infectious complications related to *B. lanceolatus* bite.	170 patients	*Bothrops lanceolatus*	*A. hydrophila* *M. morganii* *K. pneumoniae* *Bacillus* sp. *Enterococcus* sp.	Cephalosporins, aminoglycoside, ciprofloxacin, and metronidazole	Wound infection occurred in 12% of patients Soft tissue infection occurs in patients with severe envenoming.
Wound infections of snakebites from the venomous *Protobothrops mucrosquamatus* and *Viridovipera stejnegeri* in Taiwan: Bacteriology, antibiotic susceptibility, and predicting the need for antibiotics-A BITE Study	Lin, C et al. [18]	2020	Develop a suitable tool to predict the probability of developing a snakebite wound infection.	726 patients	*Protobothrops mucrosquamatus Viridovipera stejnegeri*	*E. faecalis* *Staphylococcus* sp. *Corynebacterium* sp. *M. morganii*	Amoxicillin-clavulanate, oxacillin, cefazolin, ampicillin/sulbactam	22.5% of patients developed wound infection. Use of antibiotics according to local susceptibility 44% of patients received antibiotic prophylaxis. BITE score considers hospitalization and laboratory findings for severity. Recommendation to only give antibiotics to patients with a BITE score = 5.
Clinical features, bacteriology, and antibiotic treatment among patients with presumed *Naja* bites in Vietnam	Ngo, N et al. [35]	2020	Describe the clinical and bacteriological characteristics of local wounds in patients with presumed *Naja* bite and their antibiotic treatment.	46 patients	*Naja* spp.	*E. faecalis* *M. morganii* *Enterobacter* sp. *Proteus* sp.	Clindamycin Ciprofloxacin	80% of cases had positive bacterial wound cultures. Early antibiotic use could be indicated to prevent wound necrosis and infection.
Bacterial infection secondary to *Trimeresurus* species bites: A retrospective cohort study in a university hospital in Bangkok	Kriengkrairut, S Othong, R [36]	2021	Determine the infection rate in those bitten by *Trimeresurus* spp.	123 patients	*Trimeresurus* spp.	No bacteria were identified in those infected	Antibiotics used are not specified	6.5% developed a bacterial wound infection presented as cellulitis, tenosynovitis, or necrotizing fasciitis. Antibiotic prescription rate was 12.2%. Hemorrhagic manifestations were found to be the only risk factor for infection.
Wound infection from Taiwan Cobra (*Naja atra*) Bites: Determining bacteriology, ATB susceptibility, and use of antibiotics—A Cobra BITE Study	Yeh, H et al. [19]	2021	Bacteriology of infected wounds. Compare rate of infection in wounds with and without necrosis.	195 patients	*Naja atra*	*M. morganii* *E. faecalis* *Coagulase negative Staphylococcus*	Gentamicin, ceftriaxone, ciprofloxacin, or levofloxacin as monotherapy	27% of patients developed wound infection. Wound infection was more prone to occur in moderate to severe cases.
Management and prognosis of snake envenomation among pediatric patients: A national database study	Chiang, L et al. [37]	2022	Investigate the epidemiology, management process, and endpoints of pediatric snakebite envenomation in Taiwan.	106 patients	Snake species are not specified	No bacteria were identified in those infected	Antibiotics used are not specified	65% of patients received antibiotics, but only 35% were hospitalized. Use of prophylactic antibiotics ranged from 15 to 100%. Antibiotic prophylaxis and treatment are controversial.
Secondary infection profile after snakebite treated at a tertiary referral center in the Brazilian Amazon	Mendes, V et al. [20]	2022	Characterize local secondary infections from snakebites.	545 patients	*Bothrops* spp.	*P. mirabilis* *Morganella* spp. *E. coli* *Streptococcus* sp. *Aeromonas* spp. *S. aureus* *Clostridium* spp.	Ceftriaxone, piperacillin-tazobactam, ciprofloxacin	49% of patients were diagnosed with moderate envenomation and 23% developed secondary bacterial infection. Secondary infection occurred in those with moderate envenoming.
Characteristics of snakebite-related infection in French Guiana	Houcke, S et al. [38]	2022	Assess the prevalence of wound infection after snakebite envenoming to identify the involved bacteria and monitor the use of antibiotics.	172 patients	*Bothrops* spp.	*A. hydrophila* *M. morganii* *S. aureus* *P. rettgeri*	Amoxicillin/clavulanate Cefotaxime	36% received antibiotics at admission. 32% developed wound infection: 19% had grade 1 envenoming, 35% grade 2, and 53% grade 3. 83% of isolates were resistant to amox/clav, so administration is not advised. Favor the promotion of proper use of antibiotics.
Bacteriological Studies of Venomous Snakebite Wounds in Hangzhou, Southeast China	Hu, S et al. [39]	2022	Define the pattern of wound bacterial flora of venomous snakebites and their susceptibility to common antibiotics	331 patients	*D. acutus*, *G. brevicaudus*, *T. stejnegeri*, *N. atra*	*M. morganii* *S. aureus* *A. hydrophila*	Aminoglycosides Cephalosporins Quinolones	25% developed wound infection. 23% developed localized tissue necrosis. 13% had positive bacterial cultures.
The effectiveness of antibiotics in managing bacterial infections on bite sites following snakebite envenomation	Senthilkumaran, S et al. [23]	2023	Document the bacterial profiles in local bite sites and provide guidance for the effective management of infections.	266 patients	*Daboia russelii*	*S. aureus* *Klebsiella* sp. *E. coli* *P. aeruginosa*	Linezolid Amikacin Clindamycin Piperacillin-tazobactam Colistin	82% of patients had a bacterial growth identified. Use of antibiotics as preventive measure is controversial; they should only be used when there is confirmation of a wound infection. Overuse of antibiotics can contribute to antimicrobial resistance.
Epidemiology of secondary infection after snakebites in center-west Brazil	Soares Coriolano Coutinho, J et al. [40]	2023	Evaluate incidence of secondary infections, characterize microbiological profile and empirical therapy failure rates.	326 patients	*Bothrops* spp.	*A. hydrophila*	Amoxicillin/clavulanate TMP-SMX	47.5% developed secondary infection. Only 7 had microbiological cultures. Association between infection and severity of envenomation.

## Data Availability

No original data were generated in this study.

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
