# Peer review of "Use of Antibiotics following Snakebite in the Era of Antimicrobial Stewardship"

_toxins, 2024, doi:10.3390/toxins16010037_

Round 1

Reviewer 1 Report

Comments and Suggestions for Authors

The manuscript reviews antibiotic use (prophylactic or therapeutic) to treat snakebites. For this, the authors searched only one database (PubMed) and selected 24 papers on the issue. The conclusions pointed to the use of antibiotics only in cases of infection.

The paper shows good writing and fits into Toxins.

The authors should search for more than one database for the review. Maybe they could find more eligible papers for the review and strengthen the conclusion.

Author Response

REVIEWER 1

The authors should search for more than one database for the review. Maybe they could find more eligible papers for the review and strengthen the conclusion.

Response: Thank you. Review was expanded and Scopus, EMBASE, and Cochrane library findings were included in methods (lines 187-191).

Reviewer 2 Report

Comments and Suggestions for Authors

The manuscript describes the results of a literature review on the use of antibiotics to treat post-snake bit poisoning infections. The subject is controversial, and the work shows its use in an interesting and profound way. The work is well written with necessary tables and references. We suggest your publication.

Minor points

1) Table 1: Remove the italics from sp. and spp.

2) Correct the name of some antibiotics that were cut due to space in the table.

Comments on the Quality of English Language

English is fine, minor mistakes can be resolved during editing.

Author Response

REVIEWER 2

Table 1: Remove the italics from sp. and spp

Response: Changes were made in the table.

Correct the name of some antibiotics that were cut due to space in the table.

Response: Changes were made in table 1.

English is fine, minor mistakes can be resolved during editing

Response: The whole document was reviewed again, and corrections were made.

Reviewer 3 Report

Comments and Suggestions for Authors

Reference:  “Use of Antibiotics Following Snakebite in the Era of Antimicrobial Stewardship”. Submitted to TOXINS, November, 2023.

General comments: In the manuscript submitted for this evaluation, the authors present a literature review of 24 scientific publications, where they discuss the use of antibiotics in the treatment of accidents involving snake bites. Throughout the text, literature data is presented indicating the use of antibiotics in the prophylactic treatment of bacterial infections in snakebite victims based on the presence of bacteria in the fangs or snake oral cavity or in the tissues exposed. Throughout the text, the various antibiotics used are discussed, as well as the reasons why those responsible for treatments chose to use antibiotic therapy with or without bacterial infections. Finally, the authors give their opinions on antibiotic therapy in the treatment of accidents caused by snakes, suggesting that this practice should be restricted to patients at high risk of developing bacterial infections or those who already have bacterial infections. This is a lean, direct and well-focused text. Subject falls within the scope of TOXINS and is current. The authors present a text that shows a good expertise of the subject that deserves to be published, however, I suggest changes that can make the revised text more complete and attractive.         

Specific Comments:

1-    Between lines 38 and 39 the authors wrote…Bacterial infections are a secondary complication of wounds caused by animal bites, including those inflicted by snakes. Here authors could indicate some references.   

2-    In the line 68, the authors wrote …in some settings the prevalence of infections in SBE patients is lower. I think that although authors indicate two references for this assortment, they could detail more some settings along the text.

3-    In the table 1, Included studies with main characteristics, objectives, and principal outcomes and observations. First reference, Kerrigan, K [16] 1992, Authors indicated the use of aminoglycoside, but what. There are several many different?

 4-    In the page 6, table 1, … stejnegeri in Taiwan: Bacteriology, antibiotic susceptibility, and predicting the need for antibiotics-A BITE Study. The authors did not indicate details a reference, bacteria isolated, antibiotics used, number of patients, snake species…. Please see and corrected if it is possible?

 5-    In the table 1, Included studies with main characteristics, objectives, and principal outcomes and observations. Reference, Mendes, V et al [20] 2022, Authors indicated the use of Third generation cephalosporins, but what. There are several many different?

 6-    In the page 7, table 1, … center in the Brazilian Amazon. The authors did not indicate details a reference, antibiotics used, number of patients, snake species…. Please see and corrected if it is possible?

 7-     Houcke, S et al [27] 2022, table 1, page 8. Authors indicated the use of Third generation cephalosporins, but what. There are several many different?

 8-     Table1, page 9, first line, please include reference and additional details, if possible?

 9-    Finally, table 1 is good and complete, but in my opinion, authors could organize it in a time-dependent way. They initially use references according year of publication, but along the table this good, useful and elegant procedure was not used. In a revised version authors could reorganize table 1 using references in a time-dependent manner.

 10- Between lines 75 and 76 the authors wrote … Risk factors for wound or soft tissue infection were studied in envenoming caused by diverse species of the families Viperidae and Elapidae. In my opinion a reference could be included here.

 11- Between lines 77 to 79 the authors wrote … a consistently higher inci dence of infection was described in patients with clinically moderate to severe envenoming, [22] including cases with necrosis. [27]. Necrosis which is characterized by destruction of tissue is a clinical characteristic that favors the presence of secondary bacteria infections. In my opinion, authors could dedicate a bigger space of the text to discuss the influence of snakes that cause necrosis with the presence of infections and really the clinical indication of antibiotics.

12- The statement written on pages 132 and 133 ... on the predominant bacteria of the mouth of snakes in each region… It seems difficult to control, as it would be necessary to collect snakes from the wild and do culture of their secretions, and even then, it would not be guaranteed that the microbiota in the mouths of these animals would not have significant variations.

 13- In the pages 135 and 136 … For example, in the case of aminoglycosides, their use might be detrimental in the case of neurotoxic envenoming…. And also nephrotoxic side effects caused by administration of this class of antibiotic.  

 14- The sentence wrote between lines 155 to 157 could be part of conclusion… Prospective studies need to be conducted to establish the actual incidence of infection in SBE in different settings, to correlate microbiological findings and pathology at the wound site, as well as to select the most effective antibiotic therapy.

 15- This work offers a review of the literature and provides recommendations on when to use antibiotics (pages 161 and 162). In my opinion authors should change this sentence by one more conclusive.  The use of antibiotics is not recommended for prophylactic purposes in cases of SBE. Only when there is a clear risk of bacterial infection or it has already been detected.

 16- In the line 170… A search for biomedical literature in PubMed database was carried out. Please include the electronic address of PubMed and other electronic site used through out of the text!

 17-  A good indication that I suggest the authors make in a revised version would be to separate the species of snakes involved in the accidents between those that cause tissue destruction with necrosis, such as Genus Bothrops, of others that do not cause tissue destruction, and thus perhaps find a logic in antibiotic therapy, if it exists?

 18- If it is also possible to identify patients who actually had bacterial infections after accidents, and in these cases, if available in the literature, identify characteristics that may justify antibiotic therapy.

 19-  What is the difference of information between table 1 of the main text, and that of supplementary material? It seems to me the same table?   20- Also, why did the authors submit a figure as supplementary material if it is not cited throughout the text?

Author Response

REVIEWER 3

Specific Comments: 

1-    Between lines 38 and 39 the authors wrote…Bacterial infections are a secondary complication of wounds caused by animal bites, including those inflicted by snakes. Here authors could indicate some references.   

Response: References were added. See line 39.

2-    In the line 68, the authors wrote …in some settings the prevalence of infections in SBE patients is lower. I think that although authors indicate two references for this assortment, they could detail more some settings along the text. 

Response: Following the suggestion of the reviewer, new information was added in lines 69-70

3-    In the table 1, Included studies with main characteristics, objectives, and principal outcomes and observations. First reference, Kerrigan, K [16] 1992, Authors indicated the use of aminoglycoside, but what. There are several many different? 

Response: The aminoglycoside used in this study was gentamicin. This was added in table 1.

 4-    In the page 6, table 1, … stejnegeri in Taiwan: Bacteriology, antibiotic susceptibility, and predicting the need for antibiotics-A BITE Study. The authors did not indicate details a reference, bacteria isolated, antibiotics used, number of patients, snake species…. Please see and corrected if it is possible? 

Response: The information is included in the Table in the revised version.

 5-    In the table 1, Included studies with main characteristics, objectives, and principal outcomes and observations. Reference, Mendes, V et al [20] 2022, Authors indicated the use of Third generation cephalosporins, but what. There are several many different?

Response: Ceftriaxone was the 3rd generation cephalosporin used. It was corrected in the table.

 6-    In the page 7, table 1, … center in the Brazilian Amazon. The authors did not indicate details a reference, antibiotics used, number of patients, snake species…. Please see and corrected if it is possible? 

Response: They are a continuation from previous page. With changes by year performed in the table, the information might have been lost. It is included in the revised version of the manuscript..

 7-     Houcke, S et al [27] 2022, table 1, page 8. Authors indicated the use of Third generation cephalosporins, but what. There are several many different?

Response: Cefotaxime was the 3rd generation cephalosporin used. It was corrected in the table.

 8-     Table1, page 9, first line, please include reference and additional details, if possible? 

Response: They are a continuation from previous page. With changes by year performed in the table, information may have been lost. It is now included in the revised version of the table.

 9-    Finally, table 1 is good and complete, but in my opinion, authors could organize it in a time-dependent way. They initially use references according year of publication, but along the table this good, useful and elegant procedure was not used. In a revised version authors could reorganize table 1 using references in a time-dependent manner.

Response: Thank you for the suggestion. The table was corrected and organized by year of publication.

 10- Between lines 75 and 76 the authors wrote … Risk factors for wound or soft tissue infection were studied in envenoming caused by diverse species of the families Viperidae and Elapidae. In my opinion a reference could be included here.

Response: Thank you, the reference was added in line 77

 11- Between lines 77 to 79 the authors wrote … a consistently higher incidence of infection was described in patients with clinically moderate to severe envenoming, [22] including cases with necrosis. [27]. Necrosis which is characterized by destruction of tissue is a clinical characteristic that favors the presence of secondary bacteria infections. In my opinion, authors could dedicate a bigger space of the text to discuss the influence of snakes that cause necrosis with the presence of infections and really the clinical indication of antibiotics.

Response: The text was expanded in lines 80-85, as suggested by the reviewer.

12- The statement written on pages 132 and 133 ... on the predominant bacteria of the mouth of snakes in each region… It seems difficult to control, as it would be necessary to collect snakes from the wild and do culture of their secretions, and even then, it would not be guaranteed that the microbiota in the mouths of these animals would not have significant variations.

Response: Statement in lines 139-141 is based on literature review, and several articles recommend that the use of antibiotics is directed to bacteria of the mouth of snakes, and no regular bacteria causing skin and soft tissue infection according to their experience and isolation. We deleted the phrase “in each region” in line 141.

 13- In the pages 135 and 136 … For example, in the case of aminoglycosides, their use might be detrimental in the case of neurotoxic envenoming…. And also nephrotoxic side effects caused by administration of this class of antibiotic.  

Response: A sentence mentioning possible nephrotoxic effects was added in line 145.

 14- The sentence wrote between lines 155 to 157 could be part of conclusion… Prospective studies need to be conducted to establish the actual incidence of infection in SBE in different settings, to correlate microbiological findings and pathology at the wound site, as well as to select the most effective antibiotic therapy.

Response: The sentence was changed to the conclusion section, in lines 175-178.

 15- … This work offers a review of the literature and provides recommendations on when to use antibiotics (pages 161 and 162). In my opinion authors should change this sentence by one more conclusive.  The use of antibiotics is not recommended for prophylactic purposes in cases of SBE. Only when there is a clear risk of bacterial infection or it has already been detected.

Response: A more concrete sentence was added in lines 171-175.

 16- In the line 170… A search for biomedical literature in PubMed database was carried out. Please include the electronic address of PubMed and other electronic site used through out of the text! 

Response: Thank you. An extended review was performed using additional databases and described in the methods section.

 17-  A good indication that I suggest the authors make in a revised version would be to separate the species of snakes involved in the accidents between those that cause tissue destruction with necrosis, such as Genus Bothrops, of others that do not cause tissue destruction, and thus perhaps find a logic in antibiotic therapy, if it exists?

Response: Since the study was based on publications describing infection in snakebite envenoming, most of the studies relate to envenomnings associated with tissue necrosis. The evidence on infections secondary to envenomings not causing tissue necrosis is scarce since the incidence of infection in these cases is very low. This review is focused on envenomings causing necrosis, as an increased risk factor for infection.

 18- If it is also possible to identify patients who actually had bacterial infections after accidents, and in these cases, if available in the literature, identify characteristics that may justify antibiotic therapy.

Response: The risk factors found in literature associated with infection after snakebite envenoming are mentioned in the results. We prefer to restrict our analysis to this type of accidents (snakebites) without discussing infections in other types of accidents.

 19-  What is the difference of information between table 1 of the main text, and that of supplementary material? It seems to me the same table?  

Response: Thank you for this observation. The manuscript was modified accordingly, and no supplementary material is included in the revised version. All the information is included in Table 1.

20- Also, why did the authors submit a figure as supplementary material if it is not cited throughout the text?

Response: Thank you, it was changed in the revised manuscript. No supplementary material was included.

Reviewer 4 Report

Comments and Suggestions for Authors

1. The main problem with this manuscript is it does not really provide guidance at the conclusion. The data presented are quite heterogenous and thus it might be difficult to provide definitive recommendations; however, this should be more clearly stated. 

2. If possible, consider expanding on limitations of previously reported data. In particular the selection of some antimicrobials in and of itself is a limitation (eg metronidazole monotherapy, aminoglycoside monotherapy). Why is there seemingly low usage of glycopeptides in the setting of SSTIs?  

3. On page 10 in the second paragraph describing drug classes, there should be better delineation of which class is being described. At present, readers could be lead to believe metronidazole is an aminoglycoside. 

4. It might be helpful to add a table which provides the frequency of organisms from the available data (eg Morganella spp. were found in 85% of cases). This data would also help to support selection of appropriate empiric regimen(s), if they are warranted.

5. The authors should consider indicating whether this is a narrative review or a systematic review. If the latter is the case, appropriate checklists should be utilized. 

Comments on the Quality of English Language

Sufficient. Minor typos/clarifications are warranted. 

Author Response

REVIEWER 4

  1. The main problem with this manuscript is it does not really provide guidance at the conclusion. The data presented are quite heterogenous and thus it might be difficult to provide definitive recommendations; however, this should be more clearly stated. 

Response: Thank you for this observation. Changes were made in the conclusion between lines 171-175 in order to present more conclusive recommendations.

  1. If possible, consider expanding on limitations of previously reported data. In particular the selection of some antimicrobials in and of itself is a limitation (eg. metronidazole monotherapy, aminoglycoside monotherapy). Why is there seemingly low usage of glycopeptides in the setting of SSTIs?  

Response: Limitations were added following this comment by the reviewer (see lines 164-168). The reason for the low usage of glycopeptides is unknown, but it is likely related to the existence of protocols based on other antibiotics.

  1. On page 10 in the second paragraph describing drug classes, there should be better delineation of which class is being described. At present, readers could be lead to believe metronidazole is an aminoglycoside. 

Response: A more detailed account on the antibiotics used has been included in the revised version of the manuscript (lines 98-101).

  1. It might be helpful to add a table which provides the frequency of organisms from the available data (egMorganellaspp. were found in 85% of cases). This data would also help to support selection of appropriate empiric regimen(s), if they are warranted.

Response: Since the information regarding organisms isolated is so heterogeneous, and populations used in each article differ so much among subjects included; a table with this information may not be accurate for every clinical setting and may lead to incorrect recommendations for antibiotic use. However, a comment was added in lines: 91-93.

  1. The authors should consider indicating whether this is a narrative review or a systematic review. If the latter is the case, appropriate checklists should be utilized. 

Response: We performed a narrative review, and it was indicated in the manuscript (highlighted in the Abstract and at the end of the Introduction).

Round 2

Reviewer 4 Report

Comments and Suggestions for Authors

Comments have been appropriately addressed. 

Comments on the Quality of English Language

Minor typos which can be fixed.